# High-speed tunable microwave-rate soliton microcomb

Yang He ®[1,4], Raymond Lopez-Rios[2,4], Usman A. Javid[2], Jingwei Ling ®[1], Mingxiao Li[1], Shixin Xue[1], Kerry Vahala ®[3] & Qiang Lin[1,2] ✉

Soliton microcombs are a promising new approach for photonic-based microwave signal synthesis. To date, however, the tuning rate has been limited in microcombs. Here, we demonstrate the first microwave-rate soliton microcomb whose repetition rate can be tuned at a high speed. By integrating an electro-optic modulation element into a lithium niobate comb micro-resonator, a modulation bandwidth up to 75 MHz and a continuous frequency modulation rate up to $5.0 \times 10^{14}$ Hz/s are achieved, several orders-of-magnitude faster than existing microcomb technology. The device offers a significant bandwidth of up to tens of gigahertz for locking the repetition rate to an external microwave reference, enabling both direct injection locking and feedback locking to the comb resonator itself without involving external modulation. These features are especially useful for disciplining an optical voltage-controlled oscillator to a long-term reference and the demonstrated fast repetition rate control is expected to have a profound impact on all applications of frequency combs.

Several photonic technologies can produce coherent microwaves including optoelectronic oscillators[1], dual-frequency lasers[2], and Brillouin lasers[3]. However, the highest frequency stability microwave signals generated by any technology (electronic or optical) are derived from frequency comb technology using the principle of optical frequency division[4]. Microcombs[5] provides a powerful way to miniaturize this approach to the chip scale. The superior coherence of soliton microcombs has led to a variety of applications[5] including optical communication[6], spectroscopic sensing[7], range measurement[8–11], optical frequency synthesis[12], and neuromorphic computing[13,14]. Recently, significant interest has been attracted to develop soliton microcombs with repetition rates in the radio and microwave frequency regimes[15–23], with the potential of microwave synthesis on an integrated chip[24].

High-speed frequency tuning and modulation are crucial for the applications of coherent microwaves. However, the repetition rates of current soliton microcombs are fundamentally determined by the physical sizes of the monolithic comb resonators, which are difficult to change after the devices are made. While the geometry of a comb resonator could potentially be deformed by piezoelectric actuation, its speed would be limited by the intrinsic slow mechanical response, which was recently shown with an operation bandwidth of ~0.6 MHz[11]. To date, the continuous tuning of the repetition rate of soliton combs primarily relies upon thermal or pump frequency control that is typically limited to audio bandwidths[15–23,25,26]. Here we demonstrate a soliton microcomb whose microwave repetition rate can be continuously tuned with a record bandwidth of 75 MHz and a frequency modulation (FM) rate $> 5 \times 10^{14}$ Hz/s. Moreover, the device offers enormous bandwidth for direct locking of microwave frequency to an external reference source, free from the bandwidth constraint existing in current comb locking approaches[16–20,22,23,27]. Besides application to microwave photonics, this high-speed rate control will be useful in all microcomb applications including frequency synthesizers[12] and optical clocks[28,29].

[1]Department of Electrical and Computer Engineering, University of Rochester, Rochester, NY 14627, USA. [2]Institute of Optics, University of Rochester, Rochester, NY 14627, USA. [3]T.J. Watson Laboratory of Applied Physics, California Institute of Technology, Pasadena, California 91125, USA. [4]These authors contributed equally: Yang He, Raymond Lopez-Rios. ✉e-mail: qiang.lin@rochester.edu

## Results

### Device design

The device is an on-chip high-$Q$ lithium niobate (LN) microresonator whose dispersion is engineered for soliton comb generation. The lithium niobate platform has recently been shown to enable self-starting and bi-directional switching of soliton states with the assistance of the photorefractive effect[30,31]. LN also exhibits a strong electro-optic Pockels effect, and here it is used for rapid tuning of the soliton repetition rate, by directly integrating electro-optic tuning and modulation elements into the comb resonator. Figure 1a shows the device concept wherein electro-optic tuning and modulation elements are integrated into the comb resonator. In essence, the LN resonator functions simultaneously as a soliton comb generator and a high-speed electro-optic (EO) modulator. As shown below, this fully integrated device not only enables high-speed tuning of the soliton repetition rate but also provides a way to tightly lock the repetition rate to an external microwave reference.

Figure 1b shows the LN chip, which was fabricated on a z-cut LN-on-insulator (LNOI) wafer platform. Figure 1c shows the detailed structure of a device that consists of a microring resonator, a pulley bus waveguide, and driving electrodes (Fig. 1a). The ring resonator is designed to have a waveguide width of 2.2 μm, which yields a group velocity dispersion of about −0.035 ps²/m in the telecom band for the fundamental quasi-transverse-electric (quasi-TE) mode family that is suitable for soliton generation. The electrodes are placed along the ring resonator waveguide with an electrode-waveguide spacing of about 4 μm so as to optimize the electro-optic tuning/modulation efficiency without impacting the optical $Q$ of the resonator. The electrodes are 525 nm thick and 5 μm wide in order to support high-speed modulation. The LN microring resonator exhibits an intrinsic optical $Q$ of ~4 million. A similar optical $Q$ is obtained for resonators ranging in radius up to 1.5 mm.

### Microwave rate solitons and phase noise performance

By increasing the radius of the ring resonator from 100 to 450 μm, we are able to vary the soliton repetition rate $f_r$ from 200 to 44.84 GHz, as shown in Fig. 2a and b. Further increase of resonator size is non-trivial due to the interference of stimulated Raman scattering. LN exhibits rich Raman scattering characteristics that were recently shown to introduce self-frequency shift (SFS) on the Kerr solitons[30]. However, Raman lasing is also possible when a Stokes frequency matches a cavity resonance[31,32]. For a resonator with a radius ≥1 mm, the free-spectral range is small enough (≤20 GHz) that Raman lasing becomes unavoidable, and this perturbs soliton generation. For the z-cut resonators used here, the most significant interference comes from the Stokes wave with a Raman frequency shift of ~19 THz ($E(LO_8)$) phonon mode of LN[33,34]. To resolve this issue, the pulley bus waveguide is designed to be critically coupled at the pump wavelength around 1550 nm and over-coupled at the Raman Stokes wavelength around 1720 nm (Supplementary Information (SI), part A). As a result, Raman lasing can be suppressed.

Figure 2c and f show two examples of soliton microcombs with $f_r$ of 19.82 and 13.5 GHz, respectively. The slight spectral distortion around 1580 nm is a side effect of the bus waveguide which is designed to be under-coupled at this wavelength (see SI, part A). This side effect, however, does not impact the integrity of the Kerr solitons and their coherence is evident in the detected microwave signal shown in Fig. 2d and g. Here, both microwave signals at 19.82 and 13.5 GHz exhibit a signal-to-noise ratio >70 dB. As shown in Fig. 2e and h, the phase noise of the microwaves is about −40 dBc/Hz at 1 kHz, reaches −100 dBc/Hz at 10 kHz, and goes below −130 dBc/Hz at 3 MHz. These phase noise levels are comparable to those demonstrated recently in other on-chip soliton platforms[19,22]. The spectral bump around 4 kHz is likely due to the impact of the frequency noise of the pump laser (New Focus, TLB-6328)[19,22].

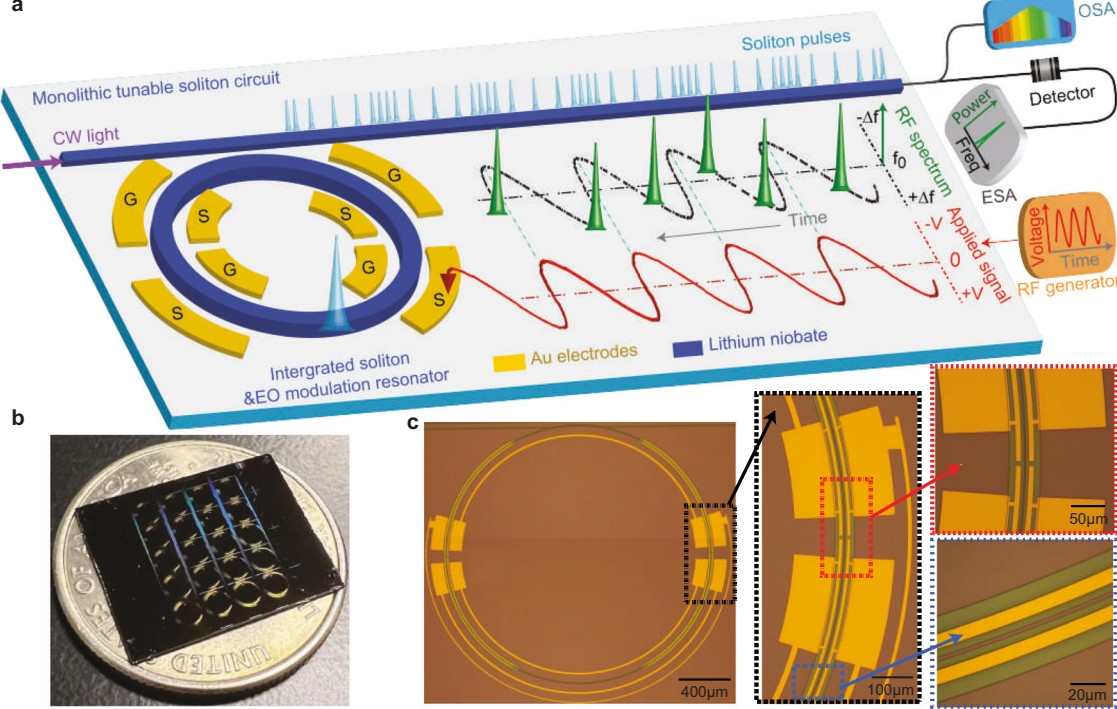

**Fig. 1 | Concept of the high-speed tunable microwave-rate soliton source.**
**a** Schematic of the tunable soliton source and its operational principle. CW continuous-wave, OSA optical spectrum analyzer, ESA electrical spectrum analyzer. **b** Photo of an LN comb resonator chip. **c** Optical images of a device and the detailed structure of the driving electrodes and resonator waveguide.

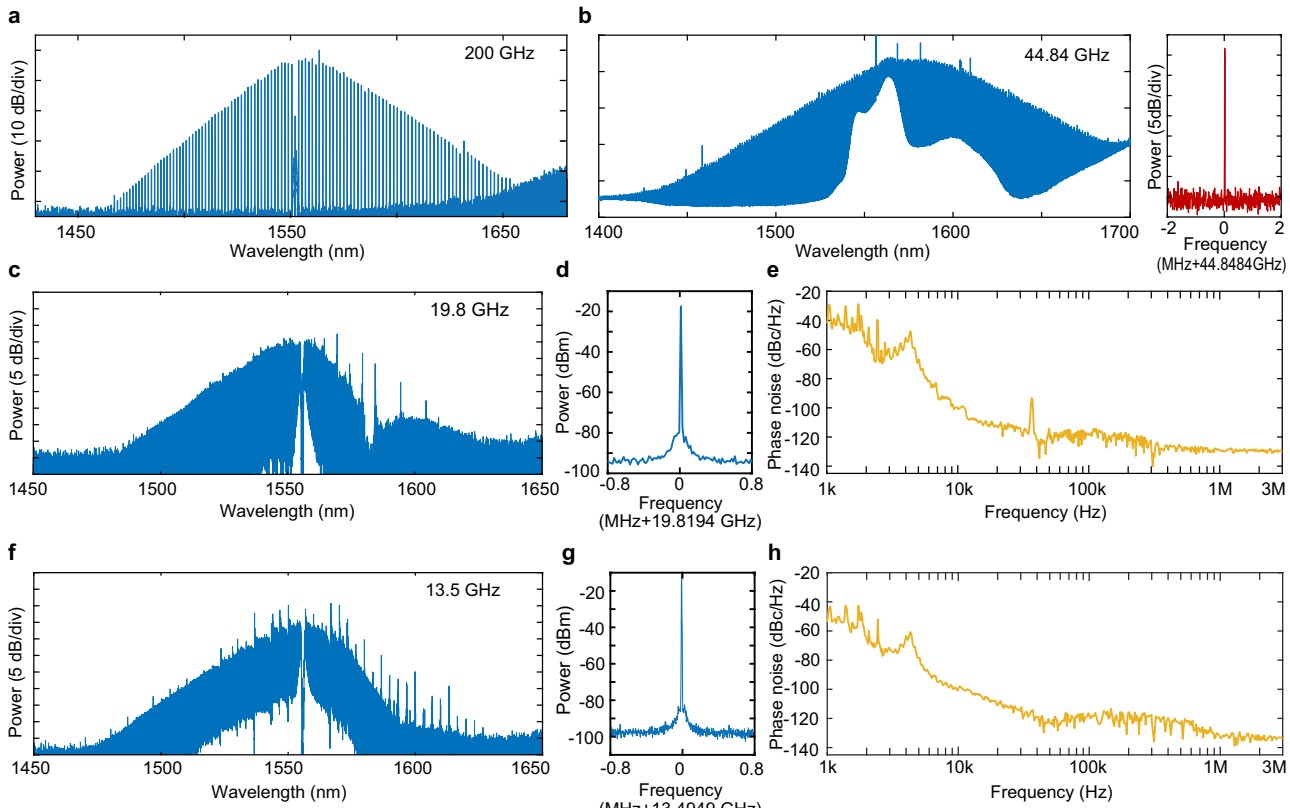

**Fig. 2 | Soliton microcombs with different repetition rates. a–c, f** Optical spectra of soliton combs with repetition rates of 200, 44.84, 19.82, and 13.5 GHz, respectively, which are produced in LN comb resonators with radii of 100 (**a**), 450 (**b**), 1020 (**c**), and 1500 μm (**f**). The corresponding on-chip pump power is 33 (**a**), 396 (**b**), 282 (**c**), and 400 mW (**f**), respectively. In **a, c, f**, the pump mode was filtered out by a fiber Bragg grating filter. The right panel in **b** shows the electrical spectrum of the detected microwave signal from the 44.84 GHz soliton comb. The resolution bandwidth (RBW) of the RF spectrum is 500 Hz. **d** and **e** Electrical spectrum and phase noise of the detected 19.82 GHz microwave signal produced by the soliton comb shown in (**c**). The soliton comb was boosted by an optical amplifier before it is detected by a high-speed detector. **g** and **h** Same as **d** and **e** but for the 13.5 GHz soliton comb shown in (**f**). In **d** and **g**, the RBW of the RF spectrum is 200 Hz. In all figures, the devices are free running without active feedback, and the pump-laser-cavity detuning is self-stabilized by the photorefractive effect.

## High-speed modulation of soliton repetition rate

In order to dynamically tune and modulate the microwave-rate solitons, we integrate EO tuning/modulating components directly onto the comb resonator as shown in Fig. 1a. For the z-cut resonator, we utilize the $r_{22}$ electro-optic tensor element of LN to tune the fundamental quasi-TE modes. For a circularly shaped ring resonator, our analysis shows that the EO tuning efficiency can be maximized with three groups of driving electrodes, each of which contains two pairs of signal-ground electrodes and spans an angle of 60° (see SI, part B for details). A detailed device layout is shown in Fig. 1c, in which only two groups of electrodes are fabricated so as to avoid interference with the coupling bus waveguide (see also Fig. 1a). To simplify experimental testing, we utilized only one group of electrodes for EO tuning and modulation. Nonetheless, the integrated EO tuning still achieves good cavity resonance tuning efficiency of 0.34 pm/V, as shown in Fig. 3a. The same figure also shows the EO-modulation response of the comb device, which gives a 3 dB bandwidth of about 61 MHz, corresponding to the photon lifetime limit of the comb resonator.

The broadband EO response of the device implies that high-speed tuning control of the soliton microcomb is possible. To show this feature, we applied a sinusoidal electric signal to the 19.81 GHz comb resonator and monitored the frequency of the detected microwave signal. As shown in Fig. 3b–e, the sinusoidal EO-drive produces a sinusoidal frequency modulation (FM) of the microwave signal. At a modulation frequency of 1 MHz (Fig. 3b), a peak driving voltage of $V_p = 0.76$ V produces an FM amplitude of 41.8 kHz which corresponds to an FM efficiency of 55 kHz/V. The FM efficiency increases considerably with the modulation frequency, reaching a value of 463 and 824 kHz/V at the modulation frequency of 10 and 50 MHz, respectively (Fig. 3c and d). As shown in Fig. 3e, we are able to modulate the microwave signal at a frequency as high as 75 MHz, where a driving voltage of $V_p = 3.0$ V produces an FM amplitude of 3.45 MHz, corresponding to an FM efficiency of 1.15 MHz/V. Here, the blurring of the time–frequency spectrum is due to the limited bandwidth (160 MHz) of the spectrum analyzer (Tektronics, RSA5126B) used for the time-dependent frequency characterization. For the same reason, the time-dependent frequency analysis likely underestimates the FM amplitude since it only captures the first-order modulation sidebands at such a high modulation frequency. Indeed, the microwave spectrum shown in Fig. 3e implies a considerably higher FM amplitude and efficiency, whose details are provided in the SI (part D). As such an FM amplitude is realized within a time scale of only ~6.7 ns (half of the modulation period), and the frequency modulation rate reaches a value of >5 × 10^{14} Hz/s at the modulation frequency of 75 MHz. The FM efficiency drops, however, with further increases in modulation frequency, due to the photon lifetime limit of the resonator. Further details are provided in SI (part E).

One mechanism responsible for the observed FM of the microwave signal is the Raman-induced SFS of the solitons whose magnitude depends on the laser-cavity detuning[35]. EO modulation of the comb resonator modulates the laser-cavity detuning of the pump wave which in turn changes the magnitude of SFS and thus shifts the carrier frequency of the Kerr solitons. Due to the group-velocity dispersion of the resonator, such a shift of soliton carrier frequency translates into a

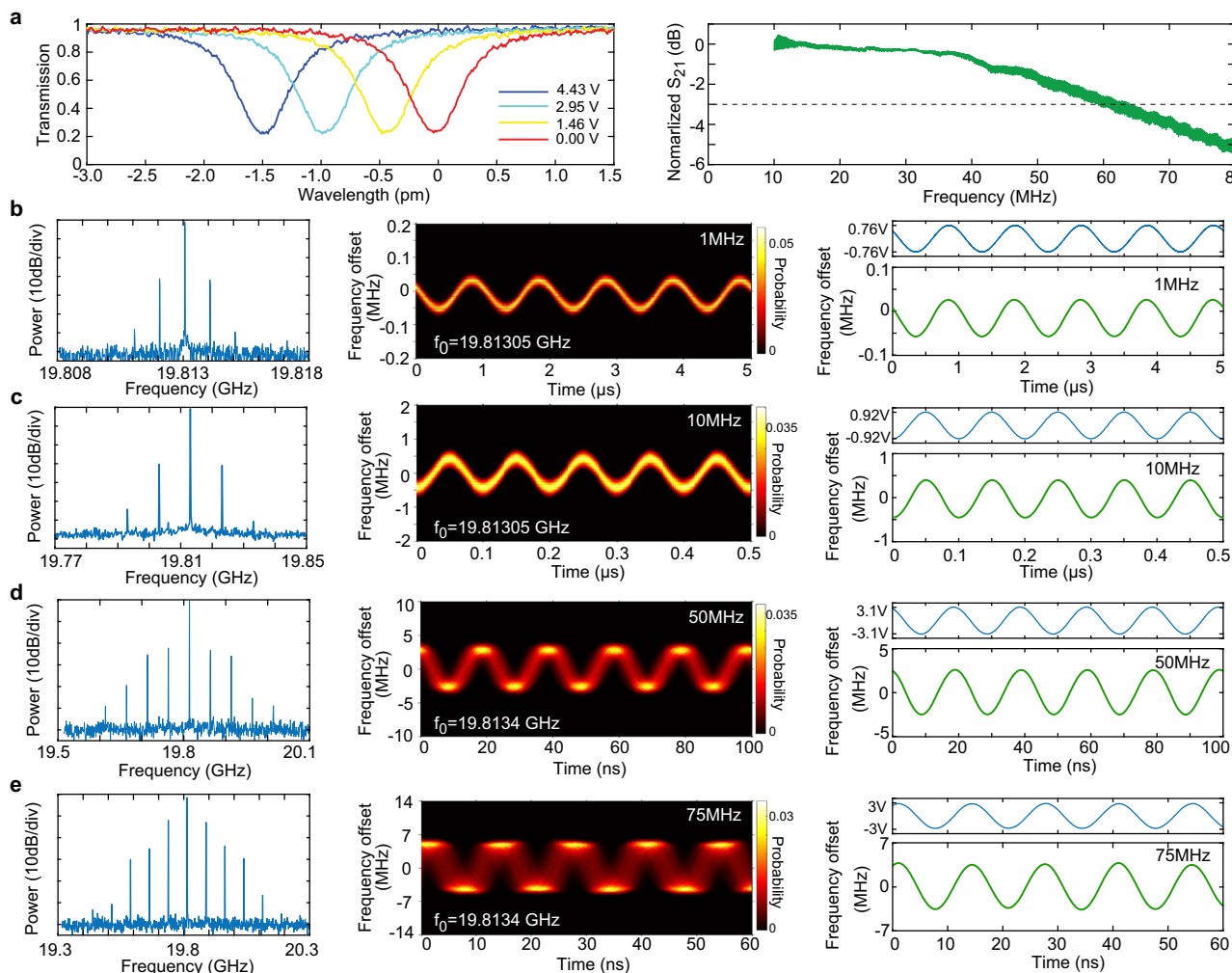

**Fig. 3 | High-speed modulation of the soliton microcombs. a** Electro-optic tuning performance of the comb resonator. Left: DC tuning of a cavity resonance. Right: Electro-optic modulation response $S_{21}$ of the comb resonator. To characterize the EO tuning performance, the pump laser power was reduced to a low level such that no comb or other nonlinear effect was produced, and the device functioned essentially as a pure electro-optic modulator. To obtain the $S_{21}$ curve, the pump laser wavelength is fixed to a cavity resonance, and the modulation amplitude of resonator transmission was recorded as a function of the modulation frequency with a network analyzer. **b–e** Frequency modulation of the soliton comb repetition rate at a modulation frequency of 1 (**b**), 10 (**c**), 50 (**d**), and 75 MHz (**e**), respectively. Left column: spectrum of the detected microwave. Center column: frequency vs. time spectrum. Right column: top: applied driving voltage; bottom: time-dependent frequency curve, which is the averaged trace of the frequency vs time spectrum shown in the center column. The data were recorded with an electrical spectrum analyzer (Tektronics, RSA5126B). In the frequency vs. time spectra are shown in the center column, the blurring of the spectrum with increased modulation frequency is due to the limited bandwidth (160 MHz) of the spectrum analyzer. In all figures, same as Fig. 2, the device is free running without active feedback, and the pump-laser-cavity detuning is self-stabilized by the photo-refractive effect.

change in the repetition rate. As shown in the SI (part C), this mechanism accounts for an FM efficiency of ~(100–200) kHz/V, which explains well the observed phenomena at low modulation frequencies. However, the FM efficiencies observed at higher modulation frequencies of 50 and 75 MHz are considerably larger than this value. The underlying reason is likely related to the speed of EO modulation which becomes comparable to the photon lifetime in the resonator so that the cavity resonances cannot adiabatically follow the EO modulation anymore. However, the exact physical mechanism is not clear at this moment and will require further exploration. On the other hand, the same mechanism of Raman-induced SFS leads to a certain extent of amplitude modulation of the produced microwave, whose details are provided in the SI (part C).

### Direct injection locking and feedback locking

The actual EO response of the device is much larger than the cavity bandwidth as evidenced in Fig. 4b. In this measurement, the resonantly enhanced EO response is apparent at the modulation frequencies of 19.81 and 39.62 GHz corresponding to one and two free-spectral ranges of the cavity. Comb-like sidebands[36,37] can be produced by driving the comb resonator at a frequency of 19.81 GHz (Fig. 4b, inset). Apparently, the device offers an EO bandwidth of up to tens of gigahertz. With this broadband modulation response of the comb resonator, we are able to lock the soliton repetition rate in two ways. On one hand, we can apply the 19.81 GHz reference microwave directly to the comb resonator during the soliton generation (Fig. 4a, yellow box). The produced EO comb then seeds the soliton generation. This approach is similar to the injection locking approach[19], but the reference microwave is now fed directly to the comb resonator itself rather than through external modulation on the pump laser. As shown in the yellow curve in Fig. 4c, such a direct locking approach is able to suppress the phase noise by about 40 dB over the frequency range of 1 Hz–1 kHz. On the other hand, we can also compare the detected microwave with the reference oscillator and apply the error signal to

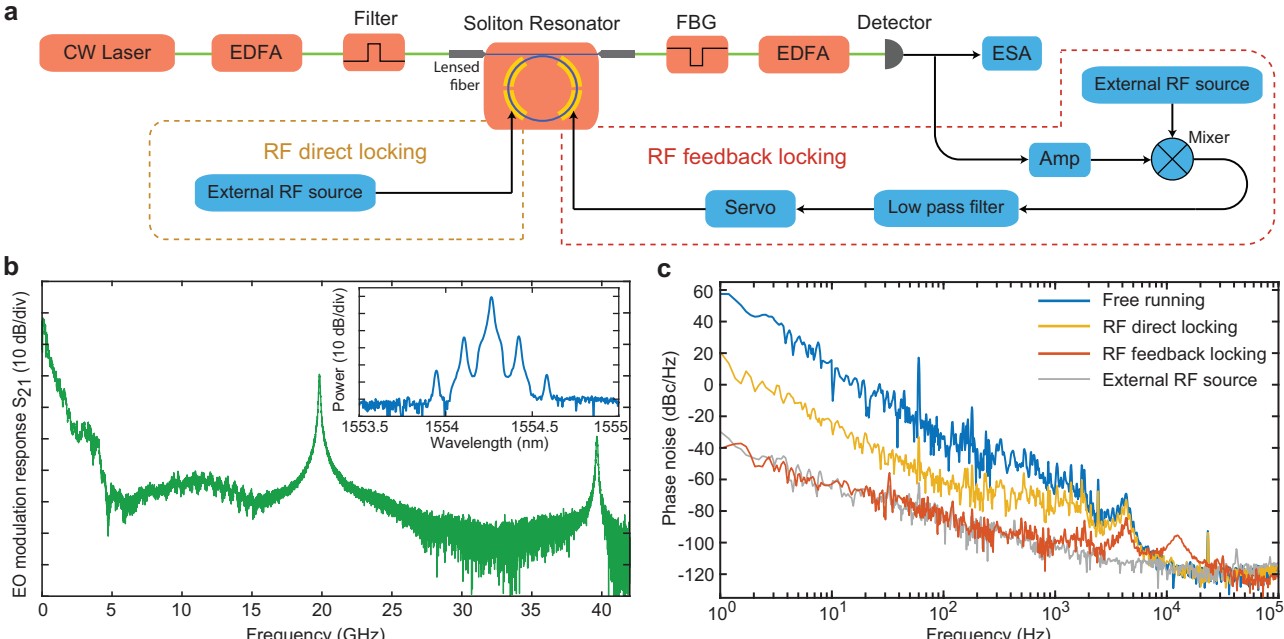

**Fig. 4 | Locking of soliton repetition rate to a reference microwave source.**
**a** Schematic of the experimental testing setup. EDFA erbium-doped fiber amplifier,
FBG fiber Bragg grating filter, Amp electrical amplifier. The soliton repetition rate is
locked to an external reference microwave source (Anritsu, MG3697C) via two
separate approaches. In the first approach, the external microwave signal directly
drives the comb resonator at a modulation rate close to the resonator FSR (yellow
dashed box). This is labeled as RF direct locking. In the second approach, the
detected soliton microwave signal is compared with the external reference
microwave, and the error signal is used to electro-optically tune the soliton repe-
tition rate (red dashed box). This is labeled as RF feedback locking. **b** Electro-optic

modulation response $S_{21}$ of the comb resonator over a broad frequency range. The
inset shows the optical spectrum of the produced comb-like sidebands when the
comb resonator was driven with a 19.81 GHz microwave signal with a power of
20 dBm. The $S_{21}$ curve was recorded at a low laser power such that the device
functions as a pure electro-optic resonator, similar to Fig. 3a. **c** Phase noise spec-
trum of the detected 19.81 GHz microwave. The yellow and red curves show the
case of RF direct locking (yellow box in **a**) and RF feedback locking (red box in **a**),
respectively, while the blue curve shows the case when the device is free running.
The gray curve shows the phase noise of the external reference microwave.

electro-optically lock the repetition rate (Fig. 4a, red box). As shown in
the red curve of Fig. 4c, this feedback-locking approach is able to
suppress the phase noise down to that of the reference microwave
over the frequency range 1 Hz–3 kHz. The residual peaks around
10 kHz are due to the bandwidth limit of the servo unit.

The current comb resonator chip is not packaged and uses lensed
fiber to couple the pump laser onto the chip (Fig. 4a). This can induce
fluctuations in the coupled pump power to the comb resonator
(especially at low frequencies). These fluctuations could be sub-
stantially reduced in the future with the packaging of the chip[19,22]. On
the other hand, the photorefractive effect of LN could have a potential
impact on the phase noise at low frequencies, whose exact behavior is
difficult to characterize at this moment and will require further
exploration in the future.

It was shown recently[38] that soliton microcombs exhibit a certain
quiet point around which the phase noise of the microwave can be
significantly suppressed due to the recoil effect between soliton SFS
and dispersive wave produced via mode crossing. This approach was
not implemented in our current experiments which focus on the basic
phase-noise performance offered by an LN soliton microcomb and the
high-speed tunability of the soliton repetition rate. The use of the
quite-point method could further improve the phase-noise perfor-
mance of the LN soliton microcombs.

## Discussion
In summary, we have demonstrated the microwave-rate soliton
microcomb whose repetition rate can be tuned at a high speed. By
taking advantage of the strong electro-optic Pockels effect of LN and
by integrating electro-optic tuning and modulation components
directly into the LN comb resonator, we are able to achieve a

continuous frequency modulation speed up to 75 MHz and a fre-
quency modulation rate up to >$5 \times 10^{14}$ Hz/s, several orders-of-
magnitude faster than existing microcomb technology[11,15–23]. The
device exhibits a modulation efficiency of >1 MHz/V, which could be
further increased if all three groups of driving electrodes are
employed. The device offers a significant bandwidth (up to tens of
gigahertz) for feedback locking of the repetition rate to an external
reference source, enabling the realization of both direct injection
locking (via intracavity EO modulation) and feedback locking (via fast
repetition-rate control), to the comb resonator itself without involving
external modulation.

The broadband locking offered by the device would enable comb-
based self-sustained coherent microwave oscillation, by self-injecting
the detected microwave to feedback lock to the comb resonator. Such
a self-sustained approach not only could improve the coherence of
microwave but also would potentially eliminate the need for an
external reference microwave source that is required in current comb-
based microwave synthesis. In addition to the microwave applications,
the flexible and high-speed temporal modulation of soliton micro-
comb may offer a new path for sensing applications such as dual-comb
spectroscopy, where the soliton comb with a time-varying repetition
rate naturally provides soliton subsets with different repetition rates
for the sensing and reference paths (with an appropriate time delay).
Such an approach would enable dual-comb or even multi-comb
spectroscopy with only a single and repetition-rate-tunable soliton
comb that could considerably simplify the sensing architecture.
Development of these applications will be left for future exploration in
the next step. Overall, the demonstrated device brings high-speed
modulation to soliton microcombs, providing a new approach to
electro-optic processing of coherent microwaves and opening up a

**Article** https://doi.org/10.1038/s41467-023-39229-3

great avenue towards high-speed control of soliton comb lines that is crucial for many applications including frequency metrology, frequency synthesis, RADAR/LiDAR, sensing, and communication.

## Methods

### Device fabrication

The devices were fabricated on a 610-nm z-cut LN-on-insulator (LNOI) wafer. Ring resonators and waveguides structures were defined by electron-beam lithography with ZEP520a as a resist, followed by etching to about 410 nm depth with Ar ion milling. After removing the ZEP520a residue, 525 nm-thick Au electrodes were patterned by electron-beam lithography with PMMA as a resist, followed by deposition using an electron beam evaporator. An overnight lift-off process was applied to remove PMMA and residual Au.

### Reporting summary

Further information on research design is available in the Nature Portfolio Reporting Summary linked to this article.

## Data availability

All data are available in the main text or the supplementary materials.

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

## Acknowledgements

We thank Qingxi Ji at Caltech for helpful discussions on RF feedback-locking experiments. This work is supported in part by the Defense Threat Reduction Agency-Joint Science and Technology Office for Chemical and Biological Defense (grant No. HDTRA11810047), the Defense Advanced Research Projects Agency (DARPA) LUMOS program under Agreement No. HR001-20-2-0044, and the National Science Foundation (NSF) (ECCS-1810169, ECCS-1842691 and, OMA-2138174). This work was performed in part at the Cornell NanoScale Facility, a member of the National Nanotechnology Coordinated Infrastructure (National Science Foundation, ECCS-1542081); and at the Cornell Center for Materials Research (National Science Foundation, Grant No. DMR-1719875). The project or effort depicted was or is sponsored by the Department of Defense, Defense Threat Reduction Agency. The content of the information does not necessarily reflect the position or the policy of the federal government, and no official endorsement should be inferred.

## Author contributions

Y.H. designed and fabricated the sample. Y.H. designed and performed the experiments. Y.H., R.L.-R., and U.A.J. did the numerical simulations. Y.H. and R.L.-R. analyzed the data. U.A.J., J.L., R.L.-R., and M.L. assisted in the experiments. S.X. assisted in device fabrication. Y.H., R.L.-R., K.V., and Q.L. wrote the manuscript. K.V. and Q.L. supervised the project. Q.L. conceived the device concept.

## Competing interests

The authors declare no competing interests.
