## [Peer review file · Nature Communications]

REVIEWER COMMENTS

Reviewer #1 (Remarks to the Author):

In the manuscript, Yang He et al. present a soliton microcomb device equipped with on-chip electrodes that enable fast electro-optic modulation of the comb repetition rate on a lithium niobate on insulator platform. The authors have achieved the fastest frequency modulation (FM) rate of $>5 \times 10^{14}$ Hz/s, and a modulation bandwidth of 75 MHz for a microcomb device, which greatly exceeds all prior attempts in the field relying on slow piezoelectric actuation. Fast tunability of the repetition rate in these devices is of large importance for many applications including frequency synthesis, LiDAR, RADAR, or telecommunications.

Overall, the paper is very well written and presented, with claims well supported by experimental data and numerical simulations. In particular, the authors demonstrate rapid FM of the repetition rate along with its stabilization to an external reference in two scenarios, which is key for lowering the timing jitter of a free-running microcomb device. The supplementary material provides a wealth of information on the waveguide and electrode design, as well as on the limitations of the modulation speed. I am certain this paper is original, new, and will be of high relevance for the microresonator, and frequency comb community. It fully complies with the standards of the Nature Communications journal.

Despite the overall excellent paper quality, I have several minor comments that may strengthen the core message of this work:

- In addition to applications that the authors have already outlined, it would be beneficial to discuss the feasibility of performing dual-comb spectroscopy with a single, repetition-rate-tunable microcomb. Because the optical tuning rate (FM rate) is on the order of hundreds of THz/s, and the repetition rate is so high, would it be possible to beat two E-fields of such a modulated source, one of which was delayed in time to produce a dual-comb (multi-heterodyne) signal? This idea is analogous to a multi-frequency FM-CW radar, but here it would be performed at multiple frequencies simultaneously because each comb line has a different optical tuning (chirp) rate. This unique feature might make this platform of high relevance for sensing.

- Does the application of a microwave field impose pure FM? Are there any losses or other mechanisms that yield amplitude modulation (AM)? A short discussion is already provided in the supplementary document, but a more elaborate description would be useful.

- Does the electrical pad capacitance impose a strong modulation frequency limit in the high-GHz regime? The authors can modulate at 20-40 GHz (far beyond the photon lifetime), but does the limitation stem from the parasitics?

Text-related comments:

- Line 53, page 1: please add a space between “bandwidths” and “[15-23]”
- Fig. 1: Please remove two consecutive periods in the caption
- Please ensure that the μ character (micro prefix) is not italicized across the manuscript.
- Reference 3: please capitalize the first letter in “brillouin”
- Reference 5/16/21: please capitalize the first letter in “kerr” – should be “Kerr”
- Reference 19: please capitalize “X”, and “K” in the “x-band/k-band”
- Reference 17: please capitalize “Q” in the “high-q”

In conclusion, I recommend publication after minor revisions.

Reviewer #2 (Remarks to the Author):

The research in this manuscript describes the design, fabrication and demonstration of a microcomb producing solitons the rate of which can be tuned at MHz speed. The microcomb is generated with a specially designed ring resonator fabricated with lithium niobate on insulator wafer platform. Special electrodes were designed for actuation of rate tuning, and the size of the microresonator and coupling waveguide was carefully designed to ensure Raman lasing did not inhibit generation of the microcomb. With this approach, the authors demonstrate generation of microwave signals at frequencies near 20 GHz and the ability to tune the output frequency at rate as large as 75 MHz.

This paper is well written and its content is technically sound. The overall design of the comb generator is novel enough, and the result of the study demonstrates achieving a tuning rate that is larger than previous studies. The generated microwave signal, nonetheless, does not have a stellar phase noise performance. The experiment also includes not-so-novel demonstration of locking the generated microwave signal at ~ 20 GHz to an external oscillator, schemes that have been previously demonstrated by other research groups. Furthermore, the issue of how the rate efficiency of modulation of the microcomb does not fit the expected analysis; the authors do not have a clear explanation.

Nevertheless, the paper in its entirety is interesting and does add to the large body of studies aimed at optical generation of reference signals for microwave applications. I recommend publication of this work after the manuscript is modified in response to the following points:

- 1- What was the laser power used for generating the output of the microcomb, and how much signal was generated at each FSR value.
- 2- In Fig. 2, c and d, there is a “hole” in the spectrum where the pump frequency is expected to be. If the pump was filtered out, it should be mentioned and the value of the pump power in the spectrum should be given.
- 3- Was there an amplifier used, or not, for Fig. 2 d and g; it would be helpful to appreciate the results if this point is clarified.
- 4- While the high rate of tuning is the major claim in this study, the only application that this reviewer can think off is using it in a VCO configuration, such as in Radar. If there are other applications, they should be specifically (rather than generally, such as metrology ...) called out. With respect to the VCO, the authors do not provide experimental evidence that tuning the resonator at any rate does not degrade the phase noise. Any additional comments in this area will be helpful.

REVIEWER COMMENTS

We thank the reviewers for their critical reading of our manuscript and making suggestions that have improved the manuscript. We have revised the manuscript wherever necessary. We list reviewers' comments in blue and our responses in black.

Reviewer #1 (Remarks to the Author):

In the manuscript, Yang He et al. present a soliton microcomb device equipped with on-chip electrodes that enable fast electro-optic modulation of the comb repetition rate on a lithium niobate on insulator platform. The authors have achieved the fastest frequency modulation (FM) rate of $>5 \times 10^{14}$ Hz/s, and a modulation bandwidth of 75 MHz for a microcomb device, which greatly exceeds all prior attempts in the field relying on slow piezoelectric actuation. Fast tunability of the repetition rate in these devices is of large importance for many applications including frequency synthesis, LiDAR, RADAR, or telecommunications.

Overall, the paper is very well written and presented, with claims well supported by experimental data and numerical simulations. In particular, the authors demonstrate rapid FM of the repetition rate along with its stabilization to an external reference in two scenarios, which is key for lowering the timing jitter of a free-running microcomb device. The supplementary material provides a wealth of information on the waveguide and electrode design, as well as on the limitations of the modulation speed. I am certain this paper is original, new, and will be of high relevance for the microresonator, and frequency comb community. It fully complies with the standards of the Nature Communications journal.

RE: We thank the reviewer for the positive comments.

Despite the overall excellent paper quality, I have several minor comments that may strengthen the core message of this work:

- In addition to applications that the authors have already outlined, it would be beneficial to discuss the feasibility of performing dual-comb spectroscopy with a single, repetition-rate-tunable microcomb. Because the optical tuning rate (FM rate) is on the order of hundreds of THz/s, and the repetition rate is so high, would it be possible to beat two E-fields of such a modulated source, one of which was delayed in time to produce a dual-comb (multi-heterodyne) signal? This idea is analogous to a multi-frequency FM-CW radar, but here it would be performed at multiple frequencies simultaneously because each comb line has a different optical tuning (chirp) rate. This unique feature might make this platform of high relevance for sensing.

RE: We thank the reviewer for this comment. Indeed, the fast tuning of soliton rep rate enables the application for dual-comb spectroscopy which we are currently investigating. We have added a few sentences in the Discussion section to discuss this potential application.

- Does the application of a microwave field impose pure FM? Are there any losses or other mechanisms that yield amplitude modulation (AM)? A short discussion is already provided in the supplementary document, but a more elaborate description would be useful.

RE: There is amplitude modulation accompanied with frequency modulation. We added a new figure (Figure S5(a)) in the Supplementary Information to better illustrate the underlying

mechanism. We also added a new reference (Ref.[8]) in the Supplementary Information, which discusses a similar mechanism.

- Does the electrical pad capacitance impose a strong modulation frequency limit in the high-GHz regime? The authors can modulate at 20-40 GHz (far beyond the photon lifetime), but does the limitation stem from the parasitics?

RE: Yes, the parasitic capacitance of the electrical pad limits the modulation efficiency at higher frequencies. It can be improved in the future by using microstructure electrodes.

Text-related comments:

- Line 53, page 1: please add a space between "bandwidths" and "[15-23]"
- Fig. 1: Please remove two consecutive periods in the caption
- Please ensure that the μ character (micro prefix) is not italicized across the manuscript.
- Reference 3: please capitalize the first letter in "brillouin"
- Reference 5/16/21: please capitalize the first letter in "kerr" – should be "Kerr"
- Reference 19: please capitalize "X", and "K" in the "x-band/k-band"
- Reference 17: please capitalize "Q" in the "high-q"

RE: We thank the reviewer for catching these typos. We have corrected them in the paper.

In conclusion, I recommend publication after minor revisions.

Reviewer #2 (Remarks to the Author):

The research in this manuscript describes the design, fabrication and demonstration of a microcomb producing solitons the rate of which can be tuned at MHz speed. The microcomb is generated with a specially designed ring resonator fabricated with lithium niobate on insulator wafer platform. Special electrodes were designed for actuation of rate tuning, and the size of the microresonator and coupling waveguide was carefully designed to ensure Raman lasing did not inhibit generation of the microcomb. With this approach, the authors demonstrate generation of microwave signals at frequencies near 20 GHz and the ability to tune the output frequency at rate as large as 75 MHz.

This paper is well written and its content is technically sound. The overall design of the comb generator is novel enough, and the result of the study demonstrates achieving a tuning rate that is larger than previous studies. The generated microwave signal, nonetheless, does not have a stellar phase noise performance. The experiment also includes not-so-novel demonstration of locking the generated microwave signal at ~ 20 GHz to an external oscillator, schemes that have been previously demonstrated by other research groups. Furthermore, the issue of how the rate efficiency of modulation of the microcomb does not fit the expected analysis; the authors do not have a clear explanation.

Nevertheless, the paper in its entirety is interesting and does add to the large body of studies aimed at optical generation of reference signals for microwave applications. I recommend publication of this work after the manuscript is modified in response to the following points:

RE: We thank the reviewer for the positive comments.

1- What was the laser power used for generating the output of the microcomb, and how much signal was generated at each FSR value.

RE: The on-chip pump power used for generating microcomb is listed in the caption of Figure 2. The output soliton combs power is about 500uW, 209uW, 12.97uW, and 54.56uW at the repetition rate of 200GHz, 44.84 GHz, 19.8 GHz, and 13.5GHz, respectively.

2- In Fig. 2, c and d, there is a "hole" in the spectrum where the pump frequency is expected to be. If the pump was filtered out, it should be mentioned and the value of the pump power in the spectrum should be given.

RE: Yes, the pump wave was filtered out by a fiber Bragg grating filter. We added a sentence in the figure caption to make it clear. This pump wave is just the residual pump wave output from the resonator, which is not very relevant to the property of the soliton comb for which the input pump power is more relevant. As such, we feel it is important to provide the input pump power (as given in the caption of Fig.2) rather than the residual output pump power.

3- Was there an amplifier used, or not, for Fig. 2 d and g; it would be helpful to appreciate the results if this point is clarified.

RE: We used an optical amplifier to boost the soliton comb power before sending it to the ESA. We have added a sentence in the figure caption to clarify this.

4- While the high rate of tuning is the major claim in this study, the only application that this reviewer can think off is using it in a VCO configuration, such as in Radar. If there are other applications, they should be specifically (rather than generally, such as metrology ...) called out. With respect to the VCO, the authors do not provide experimental evidence that tuning the resonator at any rate does not degrade the phase noise. Any additional comments in this area will be helpful.

RE: We thank the reviewer for the comments. We now added a few sentences in the Discussion Section to discuss some other potential applications such as dual-comb spectroscopy and self-injection-locked microwave oscillators.

For the phase noise during EO tuning, we expect that it is not degraded since the soliton is produced and stabilized within a few times of cavity round trips, but the EO modulation period is orders of magnitude longer. As a result, EO tuning of the resonator primarily adiabatically tunes the soliton repetition rate without disturbing its stability and thus without adding extra noises. Experimental characterization of phase noises during the fast EO tuning, however, is challenging since phase-noise characterization of high-speed frequency-modulated microwave signal requires sophisticated frequency demodulation/mixing and phase locking

(for example, see Ref. [1] below) that cannot be performed in our lab at this moment. This characterization will have to be left for future exploration.

Reference:

[1] P. Tschapek, et al, "Phase Noise Spectral Density Measurement of Broadband Frequency-Modulated Radar Signals," *IEEE Trans. Microwave Theo. Technol.* 70, 2370 (2022).

REVIEWERS' COMMENTS

Reviewer #1 (Remarks to the Author):

The authors have addressed all of my comments and made suitable changes to the manuscript. The paper is of excellent quality and meets the high quality standards of the journal. I recommend publication as is.

Reviewer #2 (Remarks to the Author):

I am happy with the changes made and recommend publication of the modified manuscript.